# ML Reproducibility Challenge 2021
# [Re] Differentiable Spatial Planning using Transformers

1  ## Reproducibility Summary

2  **Scope of Reproducibility**

3  This report covers our reproduction effort of the paper 'Differentiable Spatial Planning using Transformers' by Chaplot
4  et al. [1]. In this paper, the problem of spatial path planning in a differentiable way is considered. They show that
5  their proposed method of using Spatial Planning Transformers outperforms prior data-driven models and leverages
6  differentiable structures to learn mapping without a ground truth map simultaneously. We verify these claims by
7  reproducing their experiments and testing their method on new data. We also investigate the stability of planning
8  accuracy with maps with increased obstacle complexity. Efforts to investigate and verify the learnings of the Mapper
9  module were met with failure stemming from a paucity of computational resources and unreachable authors.

10  **Methodology**

11  The authors' source code and datasets are not open-source yet. Hence, we reproduce the original experiments using
12  source code written from scratch. We generate all synthetic datasets ourselves following similar parameters as described
13  in the paper. Training the mapper module required loading our synthetic dataset over 1.6 TB in size, which could not be
14  completed.

15  **Results**

16  We reproduced the accuracy for the SPT planner module to within 14.7% of reported value, which, while outperforming
17  the baselines [3] [5] in select cases, fails to support the paper's conclusion that it outperforms the baselines. However,
18  we achieve a similar drop-off in accuracy in percentage points over different model settings. We suspect that the
19  vagueness in the accuracy metric leads to the absolute difference of 14.7% despite the paper being reproducible. We
20  further improve the reproduced figures by increasing model complexity. The Mapper module's accuracy could not be
21  tested.

22  **What was easy**

23  Model architecture and training details were enough to easily reproduce.

24  **What was difficult**

25  We lost significant time in generating all synthetic datasets, especially the dataset for the Mapper module that required
26  us to set up the Habitat Simulator and API [4]. The ImageExtractor API was broken, and workarounds had to be
27  implemented. The final dataset approached 1.6 TB in size, and we could not arrange enough computational resources
28  and expertise to handle the GPU training. Furthermore, the description of the action prediction accuracy metric used is
29  vague and could be one of the possible reasons behind the non-reproducibility of the results.

30  **Communication with original authors**

31  The authors of the paper could not be reached even after multiple attempts.

# 1 Introduction

In the original paper [1], the problem of spatial path planning in a differentiable way is considered. The authors show that their proposed method of using Spatial Planning Transformers outperforms prior data-driven models that propagate information locally via convolutional structure in an iterative manner. Their proposed model also allows seamless generalisation to out-of-distribution maps and goals and simultaneously leverages differentiable structures to learn mapping without a ground truth map.

# 2 Scope of reproducibility

We seek to investigate the following major claims made in the paper:

- **Claim 1**:
  Their proposed SPT planner module provides a definite improvement of 7-19% over state-of-the-art CNN based planning baselines in average action prediction accuracy.

- **Claim 2**:
  Their proposed SPT planner module maintains stability in accuracy as complexity increases and the number of obstacles increases.

- **Claim 3**:
  Their proposed SPT module outperforms classical mapping and planning baselines under an end-to-end mapping and planning setting.

# 3 Methodology

The entire codebase is written from scratch for the SPT modules and the synthetic dataset generation in Python 3.6. Pytorch Lightning was used for the SPT modules. For dataset generation, similar parameters were used, as mentioned in the paper, to the maximum extent. The vagueness of parameters in terms of obstacle size allowed us to test out a range of obstacle sizes and the accuracy of the model on them. All runs were logged on the WandB platform. The training was done using NVIDIA Tesla T4 and P10 GPUs on Google Colaboratory Pro.

## 3.1 Model descriptions

Our implementation of the model follows the description provided in the paper taking liberties where details are vague. The input map and the goal map are stacked vertically and then fed into a CNN Encoder. The Encoder has 2 fully connected layers with a kernel size=1 and ReLU activation function. The first layer increases the number of channels from 2 to 64, while the second layer maintains the number of channels and outputs a 64 channel encoded input. As described in the original paper, Positional encoding is added to the encoded input, which is then reshaped and fed into the Encoder part. Their are five encoder layers, each with $n_{heads} = 8$, $d_{model} = 512$ and $dropout = 0.1$. This output is fed into a Decoder made of a fully connected layer. The Decoder gives one output for each cell. The output is then reshaped to regain its original map shape. We carry further investigations on how the number of layers in the CNN Encoder, $n_{heads}$ and $layers$ in the Encoder and embedding size affect the SPT Planner Module. Improvements were gained and are detailed in the Results section.

## 3.2 Datasets

### 3.2.1 The SPT Planner Module

We create 3 datasets for the SPT planner module, each with a map size = {15 30 50} and up to 5 randomly generated obstacles. The position of the goal is randomly chosen from a free-space cell. 2 different datasets are generated at map size = 15 with up to 10 and 15 obstacles, respectively. Each of these datasets has 100,000 maps for training, 5,000 for validation and 5,000 for testing.

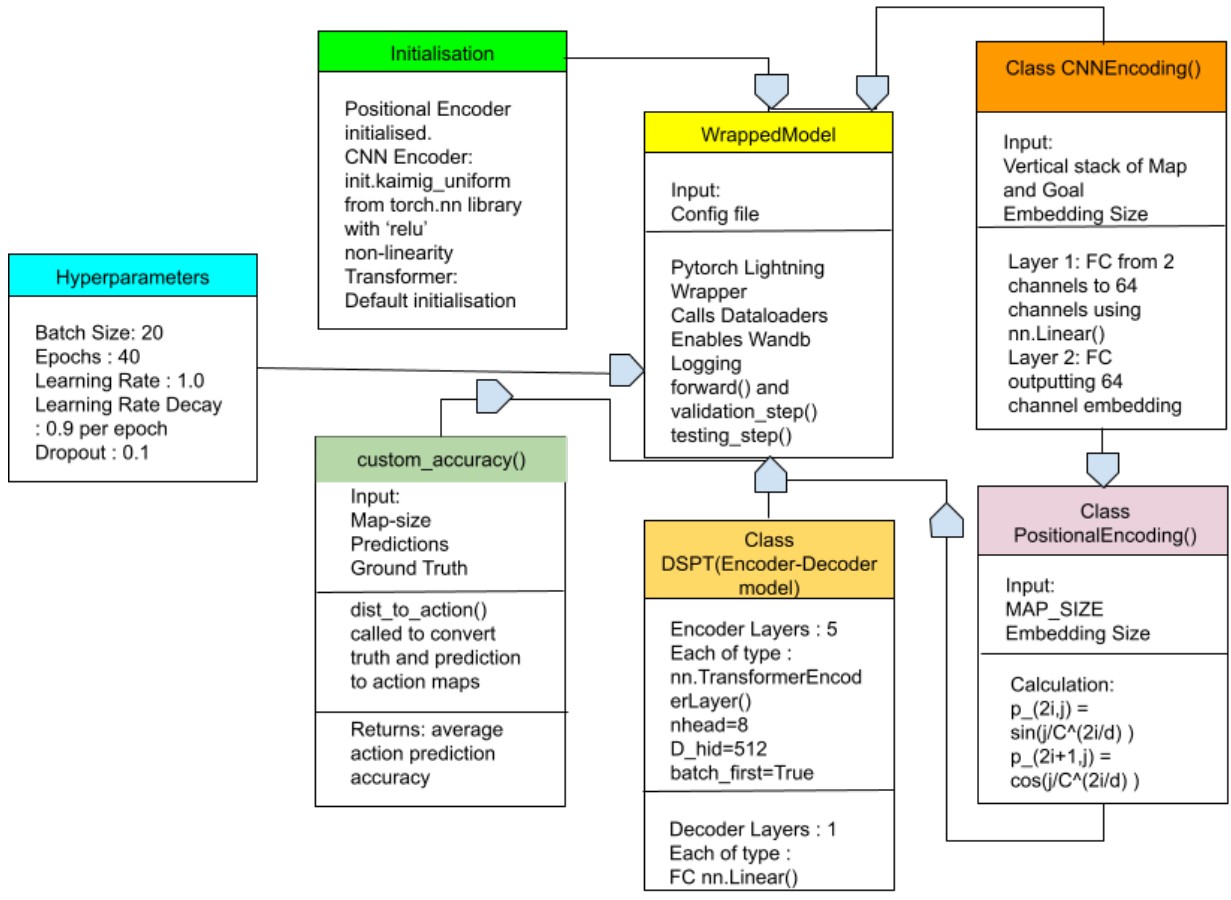

Figure 1: Code-flow diagram for our implementation.

### 3.2.2   The End-to-End Mapper and Planner Module

We further used the Habitat Simulator, and Habitat API [4] to generate 36000 maps for training the end-to-end model. Seventy-two scenes from the Gibson dataset [6] from Stanford is loaded onto the simulator, and 500 maps with a grid cell dimension of 0.5 meters and map size of 15, are rendered from each scene. Ground truths for all datasets were generated using the classical Dijkstra's algorithm. This dataset is over 1.6 TB and made it difficult to hand-engineer training on limited GPU resources.

All datasets generated and used have been released for open-source and can be found on the project's github page.

### 3.3   Hyperparameters

An extensive hyperparameter grid search led us back to the same hyperparameters cited in the paper. The model is trained for 40 epochs with a learning rate decay of 0.9 per epoch, a starting learning rate of 1.0 and a batch size of 20. The model is separately trained for each of the map distributions using mean squared error loss and stochastic gradient descent [2].

## 4   Reproducibility Results

We reproduced the accuracy for the SPT planner module to within 14.7% of reported value, which, while outperforming the baselines [3] [5] in select cases, fails to support the paper's conclusion that it outperforms the baselines. However, we achieve a similar drop-off in accuracy in percentage points over different model settings. We suspect that the

| | Navigation | | | Manipulation | | Overall |
|---|---|---|---|---|---|---|
| Method | M=15 | M=30 | M=50 | M=18 | M=36 | |
| VIN (Paper) | 86.19 | 83.62 | 80.84 | 75.06 | 74.27 | 80.00 |
| GPPN (Paper) | 97.10 | 96.17 | 91.97 | 89.06 | 87.23 | 92.31 |
| SPT (Paper) | 99.07 | 99.56 | 99.42 | 99.24 | 99.78 | 99.41 |
| **SPT (Ours)** | **84.40** | **84.83** | * | **86.49** | * | **84.74** |

Table 1: Reproducibility Results.

vagueness in the accuracy metric leads to the absolute difference of 14.7% despite the paper being reproducible. The Mapper module's accuracy could not be tested.

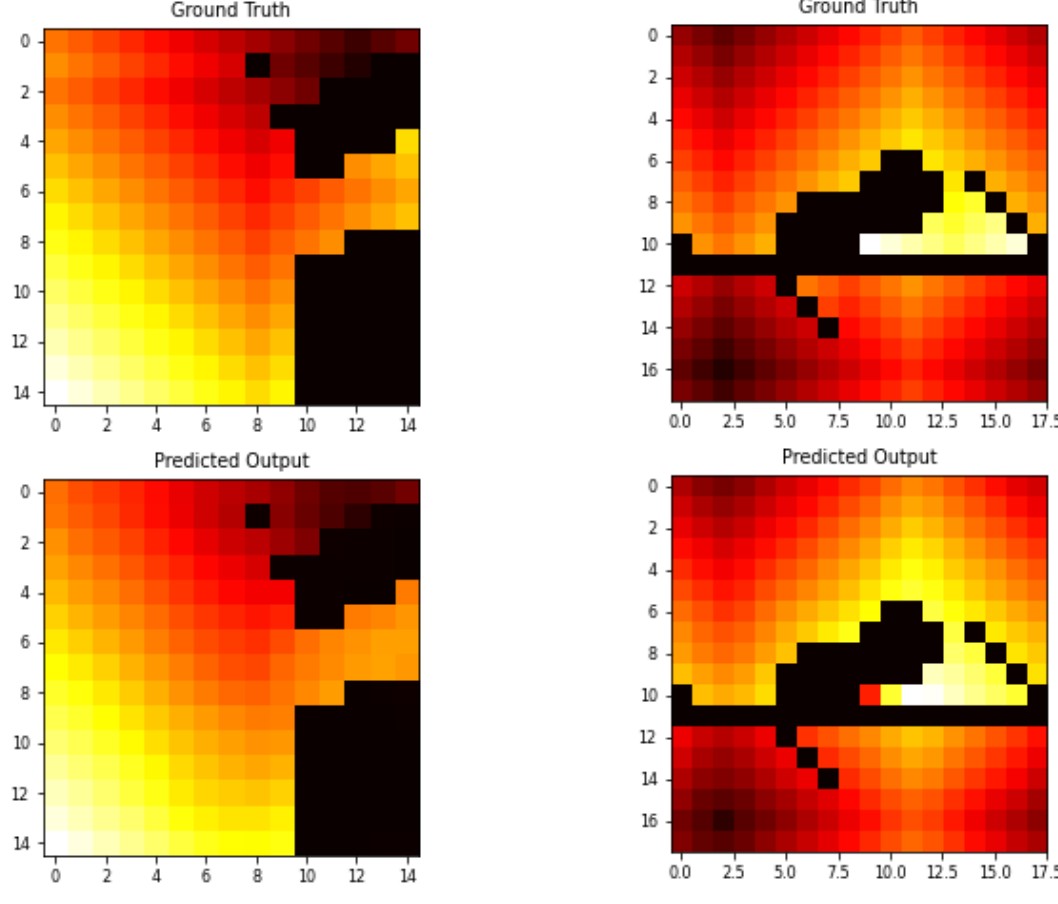

Figure 2: Accuracy : 86.71                     Figure 3: Accuracy : 83.95

Figure 4: Sample output for Navigation Task (left) and Manipulation Task (right) visualised.

* Could not be trained due to lack of enough computational resources.

# 5  Further Investigation Results and Discussion

### 5.0.1  The CNN Encoder

The CNN Encoder takes the map and the goal location as the input and encodes the information into an embedding of size $d_{model}$. This is achieved by a multi-layer, fully connected convolutional neural network. The kernel size for the convolutions is fixed at 1 to have the Encoder generate the same embedding for all input map cells. The CNN Encoder

| M=15 | | | | | |
|---|---|---|---|---|---|
| | Accuracy | Validation Loss | | Accuracy | Validation Loss |
| layers = 2 | 84.40 | 1.537 | d_model = 32 | 84.76 | 1.201 |
| layers = 4 | 84.88 | 1.166 | d_model = 64 | 84.40 | 1.537 |
| layers = 8 | **84.90** | **1.033** | d_model = 128 | **85.00** | **0.79** |

Table 2: Investigation Results on CNN Encoder parameters.

| M=15 | | |
|---|---|---|
| | Accuracy | Validation Loss |
| obstacles = N (0,5) | 84.40 | **1.537** |
| obstacles = N (0,10) | 84.31 | 2.327 |
| obstacles = N (0,15) | **84.67** | 1.614 |

Table 3: Investigation Results on increasing obstacle complexity and number.

plays a vital role in distilling the input map and representing it in the best way possible for the Transformer to act on. Table 2 lists all investigation results on the CNN Encoder parameters.

Our experiments reveal that while embedding sizes in a reasonable domain have similar accuracies, a higher embedding size provides more expressive power to the model and provides the best accuracy beating the original SPT parameters. We also see an increase in accuracy with increasing CNN Encoder layers. layers = 8 achieves the best accuracy as well as the best validation loss which shows the increase in expressive power of the encodings.

### 5.0.2 Obstacle Complexity

Obstacle complexity refers to the distribution of obstacles in the input map. The paper only cites results on input maps with a normal distribution of up to 5 obstacles. We found it crucial to test the SPT's spatial awareness and learning capabilities as this complexity is heightened. For this purpose, we created two new datasets with a higher distribution of obstacles. Table 3 lists our investigation results on these datasets.

We achieved the best accuracy on the distribution with up to 15 obstacles. However, the best validation loss is achieved with the lowest obstacles setting. This leads us to conclude that only looking at accuracy figures might be misleading because an increase in obstacles decreases the number of free spaces and consequently the number of predictions the SPT model has to generate.

### 5.0.3 The Transformer Encoder

The Transformer Encoder takes input that has been encoded into higher embedding space and has been appended with positional encoding. It is followed by a Decoder, a fully connected layer that decodes the embeddings finally given out by the Encoder. The number of multi-attention heads and encoder layers affects the expressive power of the Transformer. We conduct investigations by changing these parameters. Table 4 lists these results.

The best accuracy is achieved with $n_{heads} = 4$ and $n_{layers} = 8$. A severe drop in accuracy is found with $n_{layers} = 12$. This leads us to conclude that while increasing $n_{layers}$ increases learning capabilities of the SPT Planner module, excessive parameters might not be learnt properly from our dataset of size 100,000. The same reason suffices for an increase in $n_{heads}$.

### 5.0.4 The Best Model

The prior discussion points out that increasing the expressive power of the CNN Encoder and increasing the complexity of the Transformer Encoder helps increase the accuracy of the model. We combine all these changes to train our best model.

The parameters used are: $n_{layers} = 8$, $d_{model} = 128$, $n_{heads} = 4$ and $n_{layers} = 8$. The accuracy achieved is **85.14** with a validation loss of **0.651**. These figures beat the reproduced SPT Planner Module by **0.87%** and **57.64%** respectively.

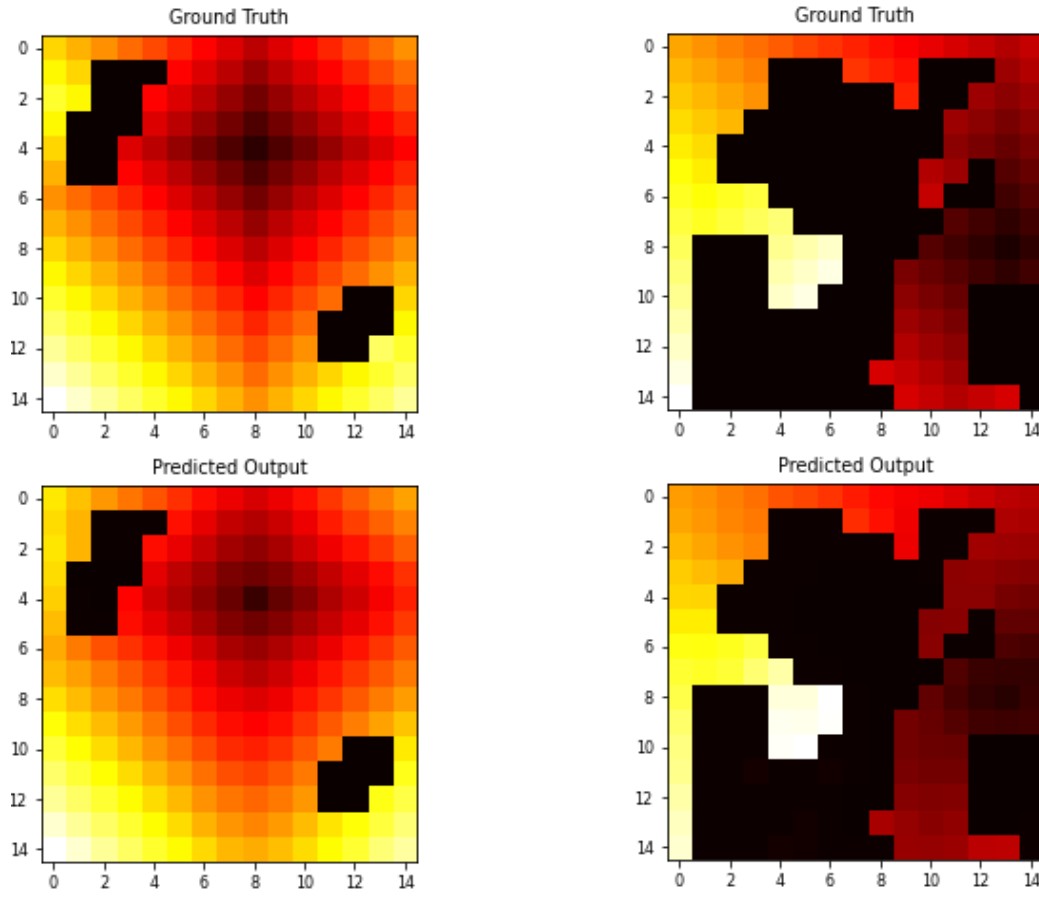

Figure 5: Accuracy : 83.49          Figure 6: Accuracy : 78.37

Figure 7: Sample output for lower obstacle distribution(left) and higher obstacle distribution (right) visualised.

| M=15 | | | | | |
| --- | --- | --- | --- | --- | --- |
| | Accuracy | Validation Loss | | Accuracy | Validation Loss |
| n_heads = 4 | **84.65** | **1.471** | n_layers = 5 | 84.40 | 1.537 |
| n_heads = 8 | 84.40 | 1.537 | n_layers = 8 | **84.96** | **1.009** |
| n_heads = 16 | 84.24 | 1.762 | n_layers = 12 | 52.33 | 40.469 |

Table 4: Investigation Results on Transformer Encoder parameters.

# 6  Discussion

## 6.1  What was easy

The easiest part of the reproduction effort was getting the Spatial Planning Transformer model up and ready from scratch. The authors' instructions regarding the layer parameters and encoder-decoder structure were abundantly clear. Furthermore, although initialisation information was missing, the model was robust enough to learn under various settings.

## 6.2  What was difficult

We lost significant time generating all synthetic datasets, especially the dataset for the mapper module that required us to set up the Habitat Simulator and API. The ImageExtractor API was broken, and workarounds had to be implemented.

The final dataset approached 1.6 TB in size, and we could not arrange enough compute resources and expertise to handle the GPU training. The SPT Planner Module could not be trained on the M=50 dataset following the same issue.

### 6.3 Reproducibility of results of SPT Planner Module

Our results lag those mentioned in the paper by a margin of over 14.7%, which makes us believe that the paper is not reproducible in its exact form. However, we achieve a similar drop-off in accuracy in percentage points over different model settings. We suspect that the paper is indeed reproducible, but the datasets' vagueness and accuracy metric lead to the exaggerated absolute difference. The lack of openly available standard datasets in the domain presents a challenge. Different papers have to report results on datasets of their choice using a metric they design themselves. The original paper's authors also did this with their synthetic datasets and a novel action prediction accuracy metric. Furthermore, these datasets are not open-sourced, and generation parameters in the paper are vague in terms of obstacle complexity and size. Our reproduction would have led to higher accuracies if the authors had provided the accuracy metric code and datasets.

Our experiments with maps of increasing obstacle complexity result in a slight increase in validation loss. This points to a plausible explanation for non-reproducibility. The non-uniformity of dataset-generation guidelines could have resulted in obstacles of greater size in our synthetic dataset.

### 6.4 Stability of the SPT Planner Module

Our results show comprehensively that the SPT Planner Module is stable concerning average action prediction accuracy for slight changes in obstacle complexity and model parameters ranging from CNN Encoder to the Transformer Encoder. This lays the ground for further research that can apply SPTs to mazes and increasingly complex scenes without considerable loss of accuracy.

### 6.5 Communication with original authors

The authors of the paper could not be reached even after multiple attempts.

## 7 Conclusion

We have tried to reproduce the paper to the best of our abilities, following the textual descriptions for source code and dataset generation to the maximum extent. We were able to improve the reproduced accuracy and loss of the SPT Planner Module by 0.87% and 57.64%, respectively, by increasing the CNN Encoder depth, embedding size and Transformer Encoder complexity. This provides ground for further research into increased complexities models that might draw deeper insights and plan more accurately.

We could not train the End-to-End Mapper and Planner Module due to a paucity of computational resources. The results that could not be reproduced are so prohibitively expensive that only very few can afford it, hence it would be better for the community if subsequent authors to this topic make their code and dataset public.

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
