# OpenReview forum: "[Re] Differentiable Spatial Planning using Transformers"
_ML_Reproducibility_Challenge/2021/Fall — RC2021_

### Official Review · Reviewer_pJA8 · 2022-03-07
**Review for reproducibility challange: Differentiable Spatial Planning using Transformers**

**Rating:** 7
**Confidence:** 3

**Review:**

The authors have made a reasonable effort in reproducing the original work, "Differentiable Spatial Planning using Transformers". The authors implemented the algorithm from scratch due to lack of open-source code from the original authors. A big challenge the authors encountered was the computational resource, which is very understandable and the relevant data that the authors share could be useful for further research in this direction. In terms of reproducing the original results, the authors did reasonable hyper-parameters tuning and also applied the method to new tasks that weren't done in the original work. These results help assess the reproducibility of the original work and thus I think it is a good contribution to the reproducibility challenge.

---

### Official Review · Reviewer_aWkc · 2022-03-07
**Reproducibility Review - Differentiable Spatial Planning using Transformers**

**Rating:** 9
**Confidence:** 2

**Review:**

The original authors make three claims and the reproducibility report sets out to verify these. The authors were unable to reproduce the results of the original paper to a high accuracy (Results were different by 14.7% which is a large margin), though they observe similar drop-off accuracy in different model settings. They state that this is possibly a result of experimenting on an entirely different dataset. The authors were being unable to contact the authors of the original paper and due to the fact that the training required large compute not all the results were verifiable. However, the results that were verified seems plausible and the structure of the paper and format are very good and easy to understand. They also attempt to visually describe their work which should be commended.

---

### Meta-Review · Area_Chair_zD4P · 2022-04-08

**Recommendation:** Accept
**Confidence:** 4

**Metareview:**

A great reproducubility study. Even though there is some discrepancy in the results presented here with that of the original paper, the paper does a good job at explaining the reason behind such behaviour which is indeed a good contribution.

---

### Decision · Program_Chairs · 2022-04-09

**Decision:**

Accept

**Comment:**

Following the recommendation of reviewers and meta-reviewer, the paper is accepted for ML Reproducibility Challenge 2021, and will be published in the upcoming special edition of ReScience Journal.